# *Streptococcus suis* Research Update: Serotype Prevalence and Antimicrobial Resistance Distribution in Swine Isolates Recovered in Spain from 2020 to 2022

**DOI:** 10.3390/vetsci11010040

**Published:** 2024-01-18

**Authors:** Máximo Petrocchi Rilo, César Bernardo Gutiérrez Martín, Vanessa Acebes Fernández, Álvaro Aguarón Turrientes, Alba González Fernández, Rubén Miguélez Pérez, Sonia Martínez Martínez

**Affiliations:** 1Animal Health Department, Veterinary Medicine Faculty, University of León, Campus de Vegazana s/n, 24071 León, Spain; mpetr@unileon.es (M.P.R.); cbgutm@unileon.es (C.B.G.M.); vacef@unileon.es (V.A.F.); algonf@unileon.es (A.G.F.); rmigp@unileon.es (R.M.P.); 2Laboratorios SYVA, Avda. De Portugal s/n, 24009 León, Spain; alvaro.aguaron@syva.es

**Keywords:** *Streptococcus suis*, swine, antibiotic resistance genes, antimicrobial susceptibility, serotype

## Abstract

**Simple Summary:**

This study aimed to update the distribution of *Streptococcus suis* serotypes in Spain, focusing on 302 clinical isolates recovered from diseased pigs between 2020 and 2022. The most prevalent serotypes were serotype 9 (21.2%), 1 (16.2%), 2 (15.6%), 3 (6%), and 7 (5.6%). Additionally, the research focused on the study of six antimicrobial resistance genes and susceptibility to 18 antimicrobials commonly utilized in pig farming. The prevalence of the antimicrobial resistance genes was the following: *tet*(O) (85.8%), *erm*(B) (65.2%), *lnu*(B) (7%), *lsa*(E) (7%), *tet*(M) (6.3%), and *mef*(A/E) (1%). Regarding antimicrobial susceptibility testing, high resistance levels were observed, particularly for clindamycin (88.4%), chlortetracycline (89.4%), and sulfadimethoxine (94.4%). The most effective antimicrobials were ampicillin (7%), gentamicin (8.9%), and ceftiofur (9.3%). Moreover, seven significant associations were identified that correlated specific antimicrobial resistance genes with observed phenotypic resistance. This research contributes to comprehending the current *S. suis* serotype distribution and ongoing antimicrobial resistance developing in Spain.

**Abstract:**

This study aimed to update the *Streptococcus suis* serotype distribution in Spain by analysing 302 clinical isolates recovered from diseased pigs between 2020 and 2022. The main objectives were to identify prevalent serotypes, differentiate specific serotypes 1, 14, 2, and 1/2, investigate specific genotypic and phenotypic antimicrobial resistance features, and explore associations between resistance genes and phenotypic resistances. Serotypes 9 (21.2%), 1 (16.2%), 2 (15.6%), 3 (6%), and 7 (5.6%) were the most prevalent, whereas serotypes 14 and 1/2 corresponded with 4.3% and 0.7% of all isolates. Antimicrobial resistance genes, including *tet*(O), *erm*(B), *lnu*(B), *lsa*(E), *tet*(M), and *mef*(A/E), were analysed, which were present in 85.8%, 65.2%, 7%, 7%, 6.3%, and 1% of the samples, respectively. Susceptibility testing for 18 antimicrobials revealed high resistance levels, particularly for clindamycin (88.4%), chlortetracycline (89.4%), and sulfadimethoxine (94.4%). Notably, seven significant associations (*p* < 0.0001) were detected, correlating specific antimicrobial resistance genes to the observed phenotypic resistance. These findings contribute to understanding the *S. suis* serotype distribution and its antibiotic resistance profiles in Spain, offering valuable insights for veterinary and public health efforts in managing *S. suis*-associated infections.

## 1. Introduction

Respiratory diseases are a prominent health concern in the porcine industry. As happens with respiratory illnesses in humans and other species, these conditions often arise due to the interplay of both primary and opportunistic infectious agents. One primary concern in pig farming arises from the Porcine Respiratory Disease Complex (PRDC), a term coined to characterize the viral and bacterial pneumonia processes that frequently manifest in growing and finishing pigs [1]. Some of the most relevant agents include the Porcine Circovirus type 2 (PCV2), the Porcine Reproductive and Respiratory Syndrome Virus (PRRSV), the Swine Influenza Virus (SIV), *Mycoplasma hyopneumoniae*, *Actinobacillus pleuropneumoniae*, *Glaesserella parasuis*, *Pasteurella multocida*, and *Streptococcus suis* [2]. In addition to the microorganisms mentioned above, non-infectious elements such as animal handling and environmental factors contribute decisively to respiratory diseases. These factors contribute substantially to the spread of pathogens, creating unfavourable conditions that lead to animal stress [3].

In these terms, *S. suis* is a major pathogen in the porcine industry [4]. This Gram-positive bacterium is part of the microbiota of pigs, as it resides asymptomatically in their upper respiratory tract, intestine, skin, and genitalia. The colonization by *S. suis* commonly occurs through close proximity with the sow and among the piglets, both during and after the farrowing process [5]. Nevertheless, upon penetration of the mucosal barriers, this microorganism enters the bloodstream, and its capsular polysaccharide (CPS) enables protection against phagocytosis and the killing mediated by monocytes and macrophages [3]. As a result, *S. suis* can survive in the bloodstream, disseminating throughout various anatomical structures. The dissemination of the bacteria may lead to conditions such as arthritis, endocarditis, meningitis, septicemia, or sudden death [6].

One of the most widespread classification techniques for *S. suis* relies on identifying specific *cps* loci, which allow the detection of up to 35 serotypes [6]. This serotype classification method is usually performed by multiplex PCR, as the one proposed by Kerdsin et al. [7]. However, this approach detects 29 capsular types, as it does not include some of the considered former *S. suis* serotypes, such as 20, 22, and 26 (reclassified to *Streptococcus parasuis*), 32 and 34 (reclassified to *Streptococcus orisratti*), and 33 (reclassified to *Streptococcus ruminantium*). Globally, the predominant capsular type is serotype 2, whereas in Europe, the predominant serotypes are 2 and 9, usually characterized by their elevated virulence [8]. Furthermore, multilocus sequence typing techniques (MLST) have permitted the complementary classification of the bacteria by their sequence type (ST), revealing the presence of more than 2500 distinctive STs [6,9].

Nevertheless, some relevant identification assays, such as the mentioned multiplex PCR, encounter limitations in distinguishing between serotype 1 and 14, as well as serotype 2 from 1/2, due to the high similarity of the clusters responsible for the expression of the capsule [7]. To overcome these limitations, in addition to traditional serological assays, alternative techniques have been proposed, such as identification by MALDI-TOF [10], rapid high-resolution melting (HRM) assays [11], PCR-Restriction Fragment Length Polymorphism methods (PCR-RFLP) [12], or mismatch amplification mutation assays (MAMA-PCR) [13]. Recently, Thu et al. achieved one of the most promising advances, presenting a straightforward and highly accurate multiplex PCR assay capable of distinguishing the mentioned serotypes in a single reaction [14]. *S. suis* typing remains a valuable diagnostic tool, providing insight into the epidemiology of outbreaks on farms and aiding in the identification of isolates more prone to cause pathogenic outcomes [15].

Despite undertaking several research projects to obtain different vaccinal antigens, researchers have not succeeded in developing a vaccine that protects against all documented *S. suis* serotypes. Additionally, the absence of efficacious commercial vaccines, paired with the early colonization of the bacteria, often resulted in pig farmers applying perinatal antimicrobials to control the disease [16]. Indeed, *S. suis* infections drive worrisome antimicrobial usage in piglets. Alarming data from worldwide surveillance on antimicrobial resistance of the bacteria emphasize the need to prioritize, as a preventive measure, restricting antimicrobial usage [17]. Moreover, *S. suis* is acknowledged as a reservoir for mobile antimicrobial resistance elements, posing a substantial risk for transmitting resistance to other pathogens [18]. Antimicrobial resistance-related infections should receive thorough attention, even in the case of *S. suis*, a concerning pathogen in the swine industry [19]. If no urgent action is taken, these infections are expected to cause ten million human deaths annually by 2050 [20].

Given the abovementioned concerns, this study aims to pursue four primary objectives: firstly, to provide an updated evaluation of the prevailing *S. suis* serotypes in Spain; secondly, to precisely detect and differentiate problematic serotypes (1, 14, 2, and 1/2) via PCR and Sanger sequencing; thirdly, to investigate specific phenotypic and genotypic antimicrobial resistance features within *S. suis*; and lastly, to determine the presence of significant associations among the examined antimicrobial resistance attributes.

## 2. Materials and Methods

### 2.1. Isolate Collection and DNA Extraction

From 2020 to 2022, a total of 302 bacterial isolates from diseased swine across various Spanish farms were analysed. Isolates were supplied by “Laboratorios SYVA” (León, Spain), with samples originating from ten different autonomous communities, and by livestock veterinarians in Castilla y León, who provided tissue samples from which *S. suis* was isolated. Among these, 128 samples were retrieved from the central nervous system (CNS), 99 from the lungs, 60 from joints, 12 from large intestines, 2 from the liver, and 1 from the spleen. For tissue samples, the isolation of *S. suis* isolates involved cauterizing the tissue surface with a spatula previously heated over a Bunsen burner. Subsequently, a cut was made with a scalpel, and a sterile swab was introduced through the incision. The swab content was then streaked onto Columbia agar plates containing 5% defibrinated sheep blood (Oxoid Ltd., Madrid, Spain), which were incubated for 24 h at 37 °C under microaerophilic conditions. The resulting colonies were translucent, small, and presented α-hemolytic activity. Nevertheless, the presumptive identification was confirmed through MALDI-TOF mass spectrometry and the *S. suis*-specific PCR that targeted the gene glutamate dehydrogenase (*gdh*).

The *S. suis* DNA extraction protocol involved resuspending approximately five colonies from Columbia agar plates in 100 µL of deionized water, followed by boiling the sample for 10 min. After centrifugation at 13,000× *g* for 10 min, the resulting supernatant was stored at −20 °C. The primers utilised for the *gdh* gene amplification were JP4 (5′-GCAGCGTATTCTGTCAAACG-3′) and JP5 (5′-CCATGGACAGATAAAGATGG-3′) [21]. The thermocycling conditions consisted of 3 min at 95 °C of preincubation, 30 cycles of 20 s at 95 °C of denaturation, 90 s at 62 °C of annealing and extension, followed by 5 min at 72 °C of final extension.

### 2.2. Molecular Typing

Four sets of multiplex PCRs were designed to identify 29 different serotypes, as described by Kerdsin et al. [7]. The first set included primers for serotypes 1 or 14, 2 or 1/2, 3, 7, 9, 11, and 16; the second covered serotypes 4, 5, 8, 12, 18, 19, 24, and 25; the third encompassed serotypes 6, 10, 13, 15, 17, 23, and 31; and the fourth targeted serotypes 21, 27, 28, 29, and 30. The DNA amplification conditions were the same as for the *gdh* gene.

For correct discrimination between serotypes 1 or 14, and 2 or 1/2, we employed the same primers as Thu et al. [14], that is, F: 5′-CTTTGTGGTGGCCTGG-3′ and R: 5′-AATGGAAGCGATGGTCAG-3. The terminal guanine of the forward primer was designed to target the single-nucleotide polymorphism (SNP) of cpsK gene at position 483. As serotypes 1 and 1/2 possess a C or a T in that position, only serotypes 2 and 14, which possess a G, will generate an amplicon. The precision of this technique facilitates the identification of problematic serotypes via the generation of 209 bp PCR bands in serotypes 2 and 14 and the absence of PCR product in serotypes 1 and 1/2.

Despite the obtained data, we designed a SNP-flanking primer pair to verify the PCR results among some of the studied isolates. The primer sequences developed were F: 5′-GGTGAATGACGGTAGTACGG-3′ and R: 5-GCAATGGAAGCGATGG-3′, which matched regions located 376 bp upstream and 196 bp downstream the SNP, respectively. Samples from the four serotypes were amplified under the following conditions: initial denaturation of 5 min at 94 °C, 35 cycles of denaturation for 30 s at 94 °C, annealing for 1 min at 53 °C and extension for 1 min at 72 °C, followed by a final extension of 5 min at 72 °C.

Following PCR amplification, the PCR products were purified using the GeneJET Gel Extraction and DNA Cleanup Micro Kit (Thermo Fisher Scientific, Madrid, Spain). After DNA quantification, Sanger sequencing was performed. Resulting sequences were aligned using MEGA 11, and SNP data were compared to prior PCR results.

### 2.3. Detection of Antimicrobial Resistance Genes

Six AMR genes were identified by multiplex PCR. The tetracycline resistance genes *tet*(O) and *tet*(M) [22] and macrolide resistance genes *erm*(B) and *mef*(A/E) [22] were detected in the same reaction, whereas lincosamide resistance genes *lsa*(E) and *lnu*(B) [23] were detected in a separate multiplex PCR. The thermocycling conditions for the first set of genes were the same as the referenced article [22] and included an initial denaturation of 3 min at 93 °C, followed by 30 cycles of 1 min of denaturation at 93 °C, 1 min of annealing at 62 °C, and 4 min of extension at 65 °C, followed by one cycle of 3 min of elongation at 65 °C. For the lincosamide resistance genes, PCR conditions were determined experimentally and included an initial denaturation of 4 min at 94 °C, followed by 30 cycles of 1 min of denaturation at 94 °C, 1 min of annealing at 51 °C, and 10 min of extension at 72 °C, followed by one cycle of 4 min of elongation at 72 °C.

### 2.4. Antimicrobial Susceptibility Testing

The antimicrobial susceptibility testing (AST) was performed using the broth microdilution method. The *S. suis* isolates were incubated in chocolate agar plates for 24 h at 37 °C under microaerophilic conditions. A McFarland turbidity standard of 0.5 was achieved by suspending *S. suis* colonies into a 5 mL of sterile saline solution. Then, 50 µL of this bacterial solution was inoculated to a 11 mL Mueller-Hinton Broth tube supplemented with 5% (*v*/*v*) fetal bovine serum. The medium was subsequently distributed by means of the Sensititre AIM™ (Thermo Fisher Scientific, Madrid, Spain) Automated Inoculation Delivery System into BOPO6F microplates, dispensing 100 µL/well. The plates were incubated for 24 h at 37 °C. 

The AST plate reading was conducted evaluating the presence or absence of precipitate in each well using Sensititre™ Manual Viewbox. The results were interpreted according to the protocols published by the Clinical Laboratory Standards Institute (CLSI) [24] and the European Committee on Antimicrobial Susceptibility Testing (EUCAST) [25]. The tested antimicrobials, their concentration ranges, and breakpoints (BP) are included in Table 1.

### 2.5. Isolate Classification in AMR Clusters

The isolates were grouped into different antimicrobial resistance (AMR) clusters depending on the corresponding phenotypic and genotypic resistance data. The statistical software SPSS version 26 (https://www.ibm.com/es-es/spss, accessed on 26 April 2023) and the data science online platform Graphext (https://www.graphext.com/, accessed on 19 January 2023) were used for this purpose. The former was needed for the initial isolate classification, whereas the later was employed for isolate reclassification and visual representation of the data. The Louvain method was the foundation for the Graphext clustering method, as it grouped related isolates into small communities and then organised them into larger ones based on their connection strength.

### 2.6. Statistical Analysis

SPSS software was also necessary to determine if statistical associations existed between the obtained data. We analysed whether there were significant associations between the presence of specific antibiotic resistance genes and the resistance observed at the phenotypic level. Furthermore, we evaluated if there were significant associations between isolates belonging to specific serotypes or anatomical locations and pertaining to specific AMR clusters. Distinct contingency tables were generated for each statistical association, and the chi-square statistic and the phi coefficient were calculated. In cases where the significance of the chi-square test was invalidated due to specific expected values in the contingency tables falling below five, Fisher’s exact test was employed.

## 3. Results

### 3.1. Molecular Typing

The combination of PCRs performed allowed the detection of 20 different capsular types. The predominant serotypes detected (representing 53% of all the isolates) were 9 (21.2%), 1 (16.2%), and 2 (15.6%). Other relevant serotypes included 3 (6.0%), 7 (5.6%), 8 (5.6%), 4 (4.6%), 1/2 (4.3%), and 19 (4%). Non-typable (NT) isolates comprised 7.6% of the total. The complete list of the detected serotypes is displayed in Table 2.

Serotype 1 or 14 and 2 or 1/2 distinction generated the following results: 96.1% of isolates positive for serotypes 1 or 14 were catalogued as serotype 1, whereas only 3.9% were serotype 14. Moreover, 78.3% of isolates positive for serotypes 2 or 1/2 were catalogued as serotype 2, whereas 21.7%% were serotype 1/2. An example of the SNP results validation via sequence alignment in MEGA 11 is displayed in Figure A2.

Serotypes 2, 1/2, 7, 9, 19, and NT isolates were primarily retrieved from the CNS. Conversely, serotypes 3, 4, and 8 were mostly collected from the lungs. Serotype 14 was the only capsular type (*n* = 2) solely isolated from the joints. Moreover, serotypes 1, 1/2, 7, and 14 were the only capsular types more isolated from the joints than from the lungs. The complete anatomical isolation site distribution patterns are shown in Table 3.

Serotypes 2 and 7 exhibited the broadest organ distribution, being isolated from five distinct anatomic locations. Conversely, serotypes 3, 8, and 14 manifested the narrowest distribution, as more than 80% of the isolates originated from a single anatomic location.

### 3.2. Detection of Antimicrobial Resistance Genes

The most frequently identified gene among the 302 isolates was the tetracycline resistance gene *tet*(O), present in 85.8% of the samples. In contrast, *tet*(M), which also confers resistance against this family of antimicrobials, was only detected in 6.3% of the isolates. A similar pattern was observed for the macrolide resistance genes. The *erm*(B) gene was identified in 65.2% of the isolates, whereas *mef*(A/E) was identified only in 1%. In the case of the lincosamide resistance genes, all isolates that amplified the *lsa*(E) gene (6.9%) also exhibited *lnu*(B) amplification (see Figure 1).

Regarding resistance patterns, *erm*(B)+/*mef*(A/E)−/*tet*(M)−/*tet*(O)+/*lsa*(E)−/*lnu*(B)− was the predominant pattern among the isolates, with 46.5% of the samples exhibiting it. The second most prevalent pattern was *erm*(B)−/*mef*(A/E)−/*tet*(M)−/*tet*(O)+/*lsa*(E)−/*lnu*(B)−, while the absence of all the resistance genes constituted the third most common pattern, observed in 9.9% of the isolates. The prevailing antimicrobial resistance patterns, both overall and within the main serotypes, are detailed in Table 4. 

### 3.3. Antimicrobial Susceptibility Testing

AST results were plotted in a heatmap (see Figure 2). The total number of antimicrobial resistances detected among all isolates was 2763 (50.7%), whereas the total number of susceptibilities encountered was 2691 (49.3%).

Based on the observed results, the most effective antimicrobials in inhibiting bacterial growth were AMP (7% resistance), GEN (8.9%), and XNL (9.3%). Conversely, the least effective antimicrobial was SDM, with a detected resistance of 94.4%.

Furthermore, only AMP and XNL presented more than 200 susceptible isolates at a specific concentration (258 and 219, respectively, at 0.25 µg/mL). Conversely, seven antimicrobials registered over 200 resistances at a specific concentration. These antimicrobials were TUL (240), TIL (241), CLI (247), TYLT (249), OXY (252), CTET (254), and SDM (285). 

For these antimicrobials, more than 70% of the isolates (79.5%, 79.8%, 81.8%, 82.5%, 83.4%, 84.1%, and 94.4%, respectively) presented CMI values superior to the highest tested concentration. Moreover, the AST of five antimicrobials revealed that 10% to 70% of the isolates exhibited CMI values above the highest tested concentration. These antimicrobials were SPE (12.9%), TIA (16.9%), NEO (26.5%), SXT (42.1%), and DANO (60.9%). The AST of six additional antimicrobials indicated that CMI values above the highest tested concentration were present in less than 10% of the isolates. These antimicrobials included FFN (0.7%), AMP (1%), GEN (1.3%), PEN (4.6%), XNL (6%), and ENRO (7.3%). Lastly, when considering all the tested isolates, 23.8% presented CMI values above the highest tested concentrations for nine or more antimicrobials.

In terms of the count of observed resistances per isolate (view Figure 3), neither absolute sensitivity nor resistance to all tested antimicrobials was detected. However, three isolates exhibited resistance to a single antimicrobial: two were resistant to SXT, while the third was resistant to NEO. Conversely, two isolates were sensitive to a single antimicrobial, with both cases involving susceptibility to FFN. Concerning the overall distribution of AMR, 65.3% of the isolates were resistant to half or more of the tested antimicrobials. The average resistance count per isolate and the median were nine, whereas the mode was 10. Finally, most of the tested isolates (80%) presented six to twelve AMR.

Following the European Medicines Agency (EMA) classification [26], 39% of the total phenotypic resistances were associated with D-category antimicrobials (SDM, CTET, OXY, SXT, PEN, SPE, and AMP), 47.9% with C-category antimicrobials (CLI, TYLT, TIL, TUL, NEO, TIA, FFN, and GEN), and 13.1% with B-category antimicrobials (DANO, ENRO, and XNL). However, this percentages are biased due to the unequal presence of D, C, and B-category antimicrobials on the BOPO6F plates (6, 7, and 3 per plate, respectively). To address this disparity, total resistance percentages per antimicrobial category were calculated, yielding the following results: 49.4% of resistances for D-category antimicrobials, 52.9% for C-category antimicrobials, and 37.9% for B-category antimicrobials.

The distribution of resistances based on the specific mechanisms of action of the antimicrobials did not reveal specific AMR patterns. Antimicrobials employing resistance mechanisms such as inhibiting folic acid synthesis (42–94%) and binding to the 30S (8–89%) or 50S (11–88%) ribosomal subunits presented broad ranges of AMR. However, antimicrobials based on impeding the synthesis of DNA gyrase subunit A (49–60%) and inhibiting the cell wall synthesis (7–21%) exhibited narrower AMR ranges.

Serotype-specific AMR distribution was also studied. Figure 4 represents the phenotypic resistance distribution patterns detected among the different capsular types. Every coloured cell in the heatmap represents the relative resistance percentage value registered for a particular antimicrobial within a specific serotype. 

Notably, AMR distribution patterns did not differ substantially between serotypes. In fact, all the tested antimicrobials presented standard deviation values among serotypes below 4%. These subtle variations can be observed in Figure 4, as antimicrobials with heterogeneous resistance levels tend to have a greater colour fluctuation among serotypes. Specifically, SXT exhibited the highest inter-serotype variation, with a 3.4% standard deviation value. PEN, DANO, ENRO, and NEO followed with 2.6%, 2.5%, 2.3%, and 2.2%, respectively. In contrast, those with low variation percentages exhibited more colour homogeneity. The antimicrobial exhibiting the lowest degree of resistance deviation was TIL, with 0.9%. 

### 3.4. Isolate Classification in AMR Clusters

Utilizing both SPSS and Graphext in tandem, isolates were grouped based on their genotypic and phenotypic AMR patterns, allowing their categorization in 11 different bacterial clusters. The main features of each cluster are shown in Table 5. However, the patterns represented in Table 5 are not the only ones present within the clusters but are the modes of the results obtained.

All AMR clusters (c1 to c11) comprised a minimum of 15 isolates, with c1, c2, and c8 exhibiting the highest sample sizes (36, 35, and 37 isolates, respectively). As for serotype distribution, 2 and 9 predominated in four distinct clusters each. In the rest of the groups, serotypes 1 (c2, c11) and 3 (c7) were the predominant capsular types. Regarding anatomical locations, the CNS was the primary bacterial isolation site in 7 of 11 clusters. The lungs were the main isolation site in the remaining groups (c2, c7, c10, and c11). 

The impact of serotypes and anatomical isolation sites on isolate clustering was minimal compared to the influence of the AMR profile. Nonetheless, 15 statistically significant associations (*p* < 0.05) were identified, correlating isolates with specific serotypes or isolation sites to specific AMR clusters. A summary of these associations is provided in Figure A1.

Genotypic AMR patterns all followed a similar distribution. In seven clusters, the main pattern was *erm*(B)+/*mef*(A/E)−/*tet*(M)−/*tet*(O)+/*lsa*(E)−/*lnu*(B). The predominant patterns in the rest of the clusters were *erm*(B)−/*mef*(A/E)−/*tet*(M)−/*tet*(O)+/*lsa*(E)−/*lnu*(B) (c8, c9, and c10) and *erm*(B)−/*mef*(A/E)−/*tet*(M)−/*tet*(O)−/*lsa*(E)−/*lnu*(B) (c11). Furthermore, all isolates presented zero to three antimicrobial resistance genes (ARG). However, apart from one isolate that belonged to c1, all samples with three ARGs belonged to c2.Isolates with two ARGs were primarily located in clusters c1 to c7, whereas one ARG isolate mainly belonged to c8, c9, and c10. Isolates with no ARGs were predominantly located in c11.

Phenotypic resistance distribution among the clusters is displayed in Figure 5, where mode values in terms of sensitivity or resistance are represented. All the clusters exhibited unique resistance patterns. Nevertheless, several common patterns were detected. Every cluster was resistant to SDM. However, the few sensitive isolates were significantly associated (*p* < 0.01) with belonging to clusters c2 and c7. The seven antimicrobial resistance pattern SXT-CTET-OXY-CLI-TYLT-TUL-TIL was present in 73.5% of the isolates, predominantly in clusters c1 to c9. Moreover, additional resistance to DANO, ENRO, and NEO was shared by almost a quarter of the isolates (24.5%). This nine AMR pattern was predominant in clusters c1, c3, and c8. 

Conversely, a minority of the isolates (5.6%) presented resistance to DANO-ENRO-NEO and were sensitive to tetracyclines, lincosamides, and macrolides. All these isolates belonged to groups c10 and c11, but contrary to what may be expected, 12 of the 17 were associated with commonly detected serotypes, such as 1 and 2. Sensitivity to FFN, AMP, and XNL was also a common pattern in every cluster. Nonetheless, the majority of the resistant isolates were grouped in c1 (*p* < 0.0001). Specific AMR distribution percentages among clusters are shown in Appendix A, Figure A1.

Isolate clustering also revealed that 9 of 11 clusters had mode values of 8 or higher for phenotypic resistances. Only c10 and c11 exhibited lower resistance patterns. The highest total AMR values were obtained in c1, as some isolates gathered 19 AMR, combining both genotypic and phenotypic resistances. Indeed, all the isolates manifesting 16 or more combined AMR belonged to c1. They were all resistant to CTET, OXY, SDM, SXT, SPE, TYLT, TUL, TIL, and CLI. However, no further associations between those samples were detected, as all belonged to six different serotypes (1, 5, 7, 9, 18, or NT) and were isolated from different anatomic locations.

Mean AMR values per cluster were also addressed, considering EMA antimicrobial categorization [26]. Cluster 1 exhibited the highest resistances per antimicrobial category, with mean AMRs per isolate of 5.2, 6.4, and 2 for D, C, and B-category antimicrobials, respectively. Clusters c4, c8, and c3 displayed the next highest mean values for D, C, and B-category antimicrobials, respectively. In contrast, c11 presented the lowest AMR count per isolate, with values of 1.3, 1, and 1.4 for categories D, C, and B, respectively. Finally, clusters c10, c7, and c11 displayed the highest relative proportions of AMR to D (52.3%), C (70%), and B (37.8%) category antimicrobials. Detailed data are available in Table A1.

Regarding AMR values and mechanisms of action, c1 demonstrated the highest overall resistance values. However, other clusters displayed higher relative resistance percentages for specific mechanisms. For example, c11 exhibited the highest AMR percentages for mechanisms based on inhibiting folic acid synthesis (31%) and DNA gyrase subunit A inhibition (33.3%). Cluster 4 presented the highest relative resistance (12%) to antimicrobials inhibiting cell wall synthesis. Lastly, c10 and c5 manifested the highest relative resistance to antimicrobials targeting the 30S and 50S ribosome subunits (47.4% and 52.9%, respectively).

The visual representation of all the clusters was depicted in Figure 6. Each isolate was represented as a node, positioned based on its unique features and correlation with other nodes. These associations between the points were represented by lines. Lines connecting these nodes represented associations between the isolates, with each node having as many lines connected to another within the same cluster as the number of shared characteristics. The colour of the isolate determined the cluster to which belonged.

This node network diagram accentuated differences in cluster homogeneity. For instance, c3, the southernmost cluster, exhibited the highest homogeneity, with closely positioned nodes. This consistency was attributed to all isolates sharing the identical AMR pattern (resistance to CTET, OXY, DANO, SDM, NEO, TYLY, TUL, TIL, CLI, ENRO, and sensitivity to the rest of the antimicrobials). Variations within this cluster originated from remaining data, as the isolates were predominantly associated with serotypes 1 and 2, were mainly obtained from the SNC, and commonly exhibited the pattern *erm*(B)+/*tet*(O)+.

In contrast, c5 was the most heterogeneous cluster. While most of the isolates exhibited a common resistance to SDM, TYLT, TUL, TIL, and CLI, the variations within this group were primarily explained by their sensitivity or resistance to tetracyclines and quinolones, including CTET, OXY, DANO, and ENRO. These specific differences are observable in Figure 7, where antimicrobial resistances among the clusters are highlighted for every tested antimicrobial. For a comprehensive understanding of the distinct resistance patterns identified among the clusters, it is advisable to analyse Figure 6 and Figure 7 simultaneously.

### 3.5. Associations between Genotypic and Phenotypic AMR

After analysing the acquired data with the corresponding statistical tools, 277 associations were detected, of which 77 (27.8%) were statistically significant (*p* < 0.05). To determine the directionality and strength of these significant associations, phi coefficients were calculated. Most of the associations (92%) manifested a positive correlation (ϕ > 0), while negative correlations (ϕ < 0) only represented 8%. Positive associations ranged from 0.15 to 0.96, whereas negative associations varied from −0.15 to −0.42. Despite the abundance of statistically significant results, the strength of most of the associations was weak (0.1 < x < 0.3). Specifically, 45.5% were weak, 33.8% moderate (0.3 < x < 0.5), and 20.8% strong (x ≥ 0.5).

Furthermore, three types of associations were detected: antimicrobial–antimicrobial (ATM–ATM), antimicrobial resistance gene–antimicrobial (ARG–ATM), and ARG–ARG. The percentages obtained for these types of associations were 58.4%, 39%, and 2.6%, respectively. The strongest ATM–ATM correlations were TYLT–TUL (0.90), TYLT–TIL (0.90), TIL–TIL (0.90), and CTET–OXY (0.90). The strongest ARG–ATM associations were *tet*(O)–TYLT (0.54), *tet*(O)–TUL (0.53), and *tet*(O)–OXY (0.50) The only significant ARG–ARG correlations observed were *erm*(B)–*tet*(O) (0.25) and *tet*(O)–*tet*(M) (−0.42).

All the significant negative associations observed were ARG–ARG or ARG–ATM. Five of them involved the *tet*(O) gene: *tet*(O)–*tet*(M) (−0.42), *tet*(O)–TIA (−0.23), *tet*(O)–XNL (−0.19), *tet*(O)–AMP (−0.16), and *tet*(O)–SPE (−0.15). Only the *lsa*(E)/*lnu*(B)–SDM (−0.23) association involved a different ARG.

The Pearson correlation coefficient was calculated for quantitative variables, such as those involving the number of genotypic or phenotypic resistances. Nevertheless, the association between those variables was moderate (0.39). Moreover, six additional correlations were observed that involved quantitative variables. The strongest ones correlated the presence of ARGs with resistance to TYLT (0.58), TUL (0.57), and TIL (0.53). All the significant associations mentioned above are displayed in Figure 8.

Furthermore, the ARGs, ATMs, and quantitative variables involved in significative associations were the following:Ten or more associations: *tet*(O) (13), CLI (11), and TIL (10);Five to ten associations: XNL (9), TYLT (9), *erm*(B) (8), TIA (8), CTET (8), OXY (8), TUL (8), no. of *ARGs* (7), AMP (7), PEN (6), SPE (6), and SXT (5);One to five associations: *lsa*(E)-*lnu*(B), GEN (4), DANO (4), NEO (4), ENRO (4), FFN (3), no. of phenotypic resistances (3), *tet*(M) (3), SDM (2), and *mef*(A/E) (1).

Although numerous significant associations concerning *tet*(O) (13), CLI (11), and TIL (10) were detected, none exhibited the highest mean ϕ values. The highest mean values corresponded to TUL ($\bar{x}$ = 0.64, s = 0.19) and TYLT (0.59, 0.24), indicating that the strongest correlations observed were associated with one of these antimicrobials. Lastly, TIA and TYLT were the antimicrobials frequently linked to the strongest associations for other variables. Specifically, correlations involving TIA or TYLT were the strongest associations detected for *tet*(M), *lsa*(E)-*lnu*(B), SPE, AMP, XNL, and *erm*(B), *tet*(O), ARGs, TUL, respectively.

Special emphasis was placed on ARG–ATM associations correlating the presence of specific ARGs with phenotypic resistances to antimicrobials from the same family. In this context, seven significant associations were detected. The associations (*p* < 0.0001) were the following:Tetracycline–ARG correlation: *tet*(O)–CTET and *tet*(O)–OXY;Macrolide–ARG correlation: *erm*(B)–TIL, *erm*(B)–TUL, and *erm*(B)–TYLT;Lincosamide–ARG correlation: *erm*(B)–CLI;Pleuromutilin–ARG correlation: *lsa*(E)–TIA.

Nonetheless, most of the associations mentioned above were moderate, falling withing the range of 0.35 to 0.49. Only the *tet*(O)–OXY association manifested a ϕ value equal to 0.50.

## 4. Discussion

This study explored the distribution of serotypes and examined phenotypic and genotypic features related to antimicrobial resistance in a collection of 302 *S. suis* clinical isolates recovered in Spain from 2020 to 2022. 

The multiplex PCR typing of the 302 bacterial isolates yielded results consistent with the prevalence observed for the different serotypes of *S. suis* in Europe over the last two decades [4,8,19,27,28]. The data obtained in previous studies [29,30,31] suggest that serotype 9 may be the predominant capsular type in Spain, followed by serotypes 1 and 2. These findings align with the ones in our isolate collection. In fact, the infection of pigs with this serotype is considered endemic in Spain [29], so its detection in more than a fifth (21.2%) of the isolates is consistent with previous findings. The relevance of serotype 9 strains should not be overlooked: although generally considered less virulent than serotype 2 strains [29], they are consistently isolated from sick pigs [32], with documented evidence of transmission to humans through exposure to raw pork products [33].

Concerning serotype 1, the second most isolated capsular type, it is one of the most prominent serotypes both nationally and across Europe [34,35]. Additionally, like serotypes 2 and 9, most of the strains associated with this serotype are isolated from sick pigs [4]. Serotype 2, the third most detected capsular type, is likely the most virulent both in animals and humans and is the predominant type globally [23,36,37]. However, it is noteworthy that strains within this serotype can exhibit high heterogeneity in virulence [37,38]. For instance, American serotype 2 strains differ from their European counterparts, primarily due to their association with different STs: European serotype 2 strains typically fall under ST 1, whereas North American strains are commonly associated with STs 25 and 28, which are less virulent [32,38].

Discriminating between serotypes 1 and 14, as well as between 2 and 1/2, has consistently posed a challenge for accurate *S. suis* serotyping. The difficulty arises from the high similarity of the clusters responsible for capsule expression, resulting in identical molecular sizes of the fragments amplified by traditional multiplex PCR (550 bp and 450 bp, respectively). Moreover, the limitations of the multiplex PCR as a diagnostic technique should be acknowledged. According to results published in Spain in 2023 [31], 43% of the isolates initially classified as serotype 2 were later identified as serotype 1/2. In contrast, only 2% of the isolates initially identified as serotype 1 were reclassified as serotype 14. It is noteworthy that serotypes 14 and 2 are frequently isolated in cases of *S. suis*-associated zoonoses, which are seldom reported [39]. Regarding serotype 1/2, it is among the most prevalent types isolated in pigs globally, following capsular types 2, 9, 1, and 3 [37]. 

Recently, Thu et al. [14] developed a novel PCR method to differentiate these serotypes within a single reaction, representing one of the most straightforward *S. suis* typing approaches published. We utilized the same primers for serotype differentiation. However, due to primer interactions with *cps1,14J* and *cps2,1/2J,* it was necessary to perform the *cps2,14K* PCR reaction in a separate master mix from the one containing primers for serotypes 1 and 2. Nevertheless, the combined results, subsequently validated through Sanger sequencing provided evidence of the sensitivity and specificity of this method, yielding serotype prevalence outcomes akin to those mentioned earlier [31].

The prevalence of the remaining serotypes was notably lower, as no capsular type was identified in more than 10% of the total isolates. Additionally, the direct involvement of minoritarian serotypes in the disease caused by *S. suis* is unclear, as clinical cases are primarily associated with the predominant capsular types [40]. In fact, some of these serotypes are typically linked to subclinical colonization of the respiratory system rather than the development of the disease [4].

Non-typable strains are generally isolated from healthy carrier pigs rather than from clinical cases [14]. These non-typable isolates may include some serotypes previously categorized as *S. suis*, non-encapsulated isolates, or new, yet uncharacterized STs that express novel capsules [41].

Regarding the distribution of AMR genes, the study from Malhotra-Kumar et al. [22] was referenced for comparison, as the primers utilized in this work derived from their study. In theirs, the prevalence values for the identified genes were as follows: *erm*(B) (56%), *tet*(M) (46.4%), *mef*(A/E) (27.2%), and *tet*(O) (8%). These percentages notably differ from the results obtained in this work, where only the *erm*(B) gene manifested similar percentages (65.2%). In contrast, *tet*(M) and *mef*(A/E) genes were observed in only 6.3% and 1% of the samples, respectively. Lastly, the *tet*(O) gene was detected in a much higher percentage than that reported in the original article, specifically in 85.8% of the isolates. It is noteworthy, however, that the original study examined a smaller number of streptococci (*n* = 125) from various species, which originated from a different country (Belgium) and period (2005).

Aside from these observations, the assertions regarding the associations between different resistance genes diverge from the findings in this study. Specifically, the proposed correlation between the presence of the *erm*(B) and *tet*(M) is questioned. This assertion was not aligned with the results obtained in this study since statistical analyses did not demonstrate a significant association between the presence of both genes (*p* = 0.28). Similarly, no significant associations were detected for the suggested correlation between the presence of the *tet*(O) gene and the *mef*(A) gene (*p* = 0.63). The only significant associations observed (both with *p* values less than 0.0001) correlated the presence of *tet*(O) with *erm*(B), and the presence of *tet*(O) with the absence of *tet*(M).

In related researches that explored the presence of those genes in *S. suis*, some were identified via PCR [23]. Nevertheless, the results differed both from those discussed in the previous article [22] and from the findings of this work. Among the twenty-one isolates they studied, all presented the *tet*(O) and *erm*(B) genes. In a similar experiment, 106 *S. suis* isolates from China were analysed [42]. The *tet*(M), *tet*(O), *erm*(B), and *mef*(A/E) genes were detected in 15.1%, 81.1%, 66%, and 48.1% of the samples, respectively. These findings align with the data acquired in this study, with the notable exception being the prevalence of the *mef*(A/E) gene, which was identified in nearly half of the isolates. Nevertheless, the study itself suggests that the detection of the *mef*(A/E) gene is uncommon.

The investigation of the *lsa*(E) and *lnu*(B) genes is infrequent compared to the ARGs mentioned above. Nonetheless, some studies have evaluated their presence in *S. suis* alongside other commonly studied genes. Bojarska et al. [23] detected both genes in one of the twenty-one isolates they analysed. Furthermore, they validated the result obtained through sequencing and verified that the *lnu*(B) gene was located downstream of the *lsa*(E) gene. Indeed, these genes might be part of genomic islands and/or clusters associated with antimicrobial resistance [43]. These findings align with the results obtained in this study, as all the isolates analysed presented either both genes or neither.

Regarding phenotypic resistances, SDM exhibited the highest resistance percentage (94.4%). However, in the case of SXT, another sulfonamide, the resistance percentage decreased to 42.1%. This increase in sensitivity may be attributed to the combination of sulfamethoxazole with trimethoprim in the BOPO6F plates (2 µg/mL of sulfamethoxazole and 38 µg/mL of trimethoprim). However, high levels of *S. suis* resistance to sulfonamides (96%) have been previously documented [44]. Recently, Hayer et al. [45] observed lower resistance levels (58.8%) to SDM. They also tested SXT, detecting only 2% of resistances. These resistance levels were lower than those reported by Arndt et al. [46], where SXT resistance increased to 13.5%. These antimicrobials are widely employed in veterinary medicine for treating bacterial infections, including those caused by *S. suis,* which may partially contribute to some of the AMR levels detected [47,48].

Tetracyclines, alongside macrolides, exhibited the most homogenous results among the various antimicrobial families tested. Specifically, the resistance percentages were 89.4% for CTET and 88.7% for OXY. Moreover, in both cases, approximately 84% of the isolates were resistant to the highest concentration assessed (8 µg/mL). In a 2012 study, [49] phenotypic resistance to tetracyclines in *S. suis* isolates (*n* = 159) was determined using antimicrobial microdilutions in plates, resulting in 83% resistance to tetracycline. Ichikawa et al. [50] recorded AMR levels of 80.7%, which were the highest for any of the antimicrobials analysed in their study. In another investigation [45], resistance levels of 95% were detected for CTET. Arndt et al. [46] recorded similar resistance levels, which reached 83.4%. Finally, Vela et al. [44] detected AMR percentages of 93.4% and 95.4% for OXY and tetracycline, respectively, in isolates from Spain. The resistance of *S. suis* isolates to tetracyclines, well documented worldwide due to their past use as growth promoters, has been recurrent for over three decades [51,52]. Together with lincosamides, tetracyclines rank among the extensively used antibiotics in the European pig industry due to their broad spectrum [53].

Concerning lincosamides, the highest resistance detected among the studies consulted was observed by Li et al. [49], with resistance percentages of 98.1%. Ichikawa et al. [50] observed lower levels of resistance among their isolates (65.8%). These values align closely with those obtained by Cantin et al. [52], who reported a resistance of 67.4%. Vela et al., in 2005, recorded AMR levels (87.4%) similar to those registered in this work (88.4%). The most divergent results (39.4%) were documented by Zhang et al. [54]. Thus, the AMR levels exhibited by this antimicrobial are variable, with a prevailing tendency toward moderate-to-high resistances. High resistance to lincosamides is also frequently associated with a phenomenon of cross-resistance to macrolides, attributed to their shared mechanisms of action. Both antimicrobial families have been extensively employed in food production animals to address several infectious diseases, which might partially explain the AMR levels observed over the past two decades [53].

The macrolide resistance results corresponded with those obtained by Vela et al. [44] (89.4% for TYLT), Li et al. [49] (84.9% for TIL), and Wang et al. [55] (90% for TIL). However, in a separate study involving isolates from the United States [56], resistances were lower (65% for TUL, 66% for TLYT, and 67% for TIL%). These values correspond closely with those reported for TIL (56.5%) by Zhang et al. [54]. Consequently, it can be inferred that resistances against this antimicrobial family tend to be high, as evidenced by the consistently elevated AMR levels observed in all the cited studies. One contributing factor might be the accessibility of these antimicrobials, given their relatively lower cost and cost-effectiveness compared to other antibiotics [57]. Furthermore, the World Health Organization has classified this antimicrobial family as “critically important” for human medicine [47]. This classification implies that these antimicrobials are indispensable for treating specific human infections, emphasizing the need for their responsible and controlled use to safeguard their effectiveness.

Regarding quinolones, the resistance levels obtained were considered intermediate since the AMR values for ENRO and DANO were 49% and 60.9%, respectively. Notably, in all studies encompassing any of these antimicrobials [44,45,47,54,55,56,58] the resistance levels were considerably lower than those observed in this study. Specifically, the resistances detected were 2%, 9.6%, 1.4%, 11.1%, 2%, 11.5%, and 1%, respectively. Therefore, when compared to the findings of this study, it is noteworthy that our results diverge remarkably from those obtained globally. However, it is important to highlight that quinolone usage constitutes 2.4% of the total antibiotic consumption in global pig farming [57]. This could, in part, contribute to the variations observed between the consulted studies and the obtained results.

The only pleuromutilin investigated in this study was TIA (16.9%). The resistance distribution for this antimicrobial mirrored that of the quinolones. Specifically, some of the AMR levels reported worldwide were 1.9% [59], 2% [56], 6.2% [45], 11.3% [54], and 19.8% [55]. Hence, it can be assumed that the AMR levels for this antimicrobial tend to be low. Indeed, resistances to pleuromutilins are predominantly identified in non-clinical isolates [47]. The only discrepant results were detected by Arndt et al. (2019) [46], who documented resistance levels of 63.7% for this antimicrobial. Pleuromutilins, alongside tetracyclines, lincosamides, and macrolides, constitute one of the antibiotic families primarily prescribed for treating respiratory and gastrointestinal infections in pigs globally [47,48]. However, a moderate increase in AMR rates to pleuromutilins has been detected in recent years [53].

The aminoglycosides studied were categorized into those with low levels of AMR (GEN with 8.9% and SPE with 12.9%) and those exhibiting moderate-to-high levels (NEO with 64.6%). The resistance values observed for GEN were generally low: 2.9% [45], 3.3% [52], and 4.6% [44]. However, higher AMR percentages were also registered, such as 30.2% [59]. Similar outcomes were also noted for SPE (38.6%) [46]. Concerning NEO, low AMR were detected, although exceptions were identified, such as the results reported by Matajira et al. (49.3%) [59] and Vela et al. (50.3%) [44]. In veterinary practice, these antibiotics are presently utilized in conjunction with β-lactams [43]. It must be noted, nonetheless, that aminoglycosides are generally associated with a naturally low level of resistance in *S. suis* [47].

With respect to amphenicols, only FFN (11.3%) was studied. Moreover, FFN was not utilized as frequently as other antimicrobials in the studies analysed. Nevertheless, it was included in some studies, obtaining 0% [56], 1% [45], and 3% [59] AMR percentages. The increased resistance to amphenicols in this study may be partially attributed to the extensive use of chloramphenicol in Spain until 1994, after which it was banned for use in animals for human consumption in the European Union [60]. Furthermore, FFN is still employed in farm and aquatic animals for treating various respiratory and digestive diseases, potentially contributing to the increasing AMR levels [61].

Concerning the resistances identified for penicillins, it is notable that they have historically been considered effective antimicrobials against *S. suis* [62,63,64]. Our study supports this observation (7% AMR levels for AMP and 21.7% for PEN). Moreover, the studies consulted demonstrated almost total sensitivity to penicillins, with AMR percentages of 0% [54], 0.1% [46], 1% [59], 0% [50], and 0% [56]. Notably, despite the global administration of β-lactams in pigs for more than five decades, most clinical *S. suis* strains continue to exhibit sensitivity to these antibiotics [43]. However, varying levels of resistance have emerged. The resistances detected were 8% [56] and 66% [49] for PEN and 19% for AMP [58]. Moreover, excluding the 66% resistance from the study from 2012 [49], the remaining AMR levels are recent, as both articles were published in 2022. According to some authors [48], the elevated resistance to penicillin is likely due to the selection pressure exerted after exposure to β-lactams used in the treatment of other diseases. 

Lastly, with regard to cephalosporins, the 9.3% resistance level to XNL aligns with the findings in other studies, as AMR levels of 0% [59], 0.7% [44], and 0.9% [45] were observed. Indeed, XNL, a third-generation cephalosporin, has proven to be the most efficacious antibiotic for addressing cases of *S. suis* in both humans and pigs until the present [65]. The only study reporting resistances over 10% was that of Arndt et al. (2019) [46], with a 56% resistance to XNL. Nonetheless, the long-term administration of this antibiotic drug may contribute to the dissemination of cephalosporin resistance in *S. suis* [65].

Clustering analysis conducted with SPSS and Graphext not only aided in categorizing the 302 isolates based on their AMR patterns but also introduced a novel visual representation approach. A notable feature was colour filtering, which adjusted the colours in the node network diagram based on the specific variables of interest. This capability enabled the precise differentiation of resistance patterns in a large number of isolates with ease (see Figure 7). To our knowledge, this is the first *S. suis* research project in which Graphext has been employed for visual representation of AMR-related data.

Regarding the 77 significant associations detected, the results should be interpreted cautiously, as the correlation between two variables does not necessarily imply a causation relation among them. In fact, both variables may be influenced by a third factor. To establish causation, a well-designed study capable of demonstrating a direct causal link is necessary, which was beyond the scope of this study. Nevertheless, considering the detected associations and observed AMR percentages, the current state of antimicrobial resistance in *S. suis* is concerning.

D-category antimicrobials are first-line treatments. However, their use should be approached with prudence and be limited to medical necessity [26]. In this context, the only viable treatment options given their low resistance levels include AMP (7%), SPE (12.9%), and PEN (21.2%). Nonetheless, 58.9% of the isolates presented an MIC of 32 µg/mL for SPE, which is the antimicrobial concentration prior to the breakpoint. Hence, the possibility of continued development of resistance against this antimicrobial cannot be ruled out.

C-category antimicrobials should be reserved for specific veterinary indications, particularly when D-category antimicrobials are unavailable due to AMR [26]. Within this category, only three antimicrobials exhibit relatively low percentages of resistance: GEN (8.9%), FFN (11.3%), and TIA (16.9%). Nevertheless, resistances may continue developing, similarly to antimicrobials of the previous category. Indeed, for GEN and FFN, 40.7% and 76.8% of the isolates, respectively, presented an MIC value equivalent to their corresponding breakpoints.

Lastly, B-category antimicrobials include those whose use in animals should be limited to mitigate the risk of AMR transmission to public health [26]. Among these, the only effective antimicrobial is XNL since it is the only one with low resistance levels (9.3%). In fact, most of the isolates (72.2%) were sensitive to the lowest concentration of the antimicrobial (0.25 µg/mL). However, as stated above, the use of such antimicrobials should be controlled, limiting their use to specific cases where alternative options are not available. Indeed, as with the rest of the available antimicrobials, their clinical utilization should consistently be justified via antimicrobial susceptibility tests.

## 5. Conclusions

This study focused partly on updating the current *Streptococcus suis* serotype distribution in Spain by typing 302 isolates, and specific serotypes 1, 14, 2, and 1/2 were distinguished by multiplex PCR. Serotypes 9 (21.2%), 1 (16.2%), 2 (15.6%), 3 (6%), and 7 (5.6%) were the most prevalent, whereas serotypes 14 and 1/2 were associated with 4.3% and 0.7% of the isolates, respectively.

The investigation of specific phenotypic and genotypic antimicrobial resistance features of the mentioned isolates were also addressed. The prevalence of six antimicrobial resistance genes analysed was the following: *tet*(O) 85.8%, *erm*(B) 65.2%, *lsa*(E) and *lnu*(B) 6.9% each, *tet*(M) 6.3%, and *mef*(A/E) 1%.

The susceptibility testing for 18 antimicrobials demonstrated high resistance values, with a mean of 10.6 resistances per isolate. The most effective antimicrobials were ampicillin (7% of resistances), gentamicin (8.9%), and ceftiofur (9.3%), whereas less efficient ones were clindamycin (88.4%), chlortetracycline (89.4%), and sulfadimethoxine (94.4%).

Lastly, seven significant associations were identified (*p* < 0.0001) that linked the presence of particular antimicrobial resistance genes with the observed phenotypic resistance to an antimicrobial within the same family.

## Figures and Tables

**Figure 1 vetsci-11-00040-f001:**
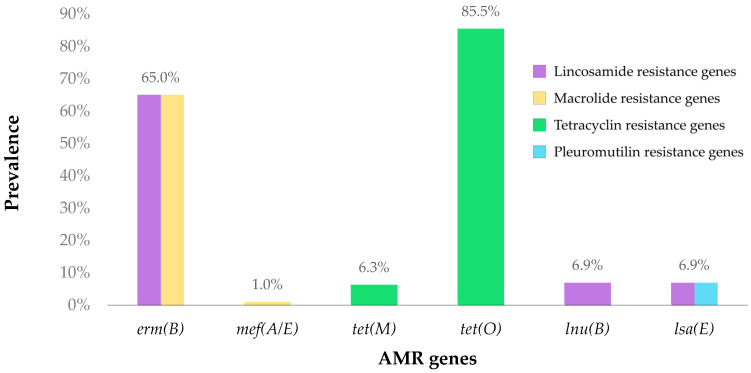
Antimicrobial resistance gene distribution of the studied isolates. Two-coloured bars represent genes that confer resistance against more than one antimicrobial family.

**Figure 2 vetsci-11-00040-f002:**
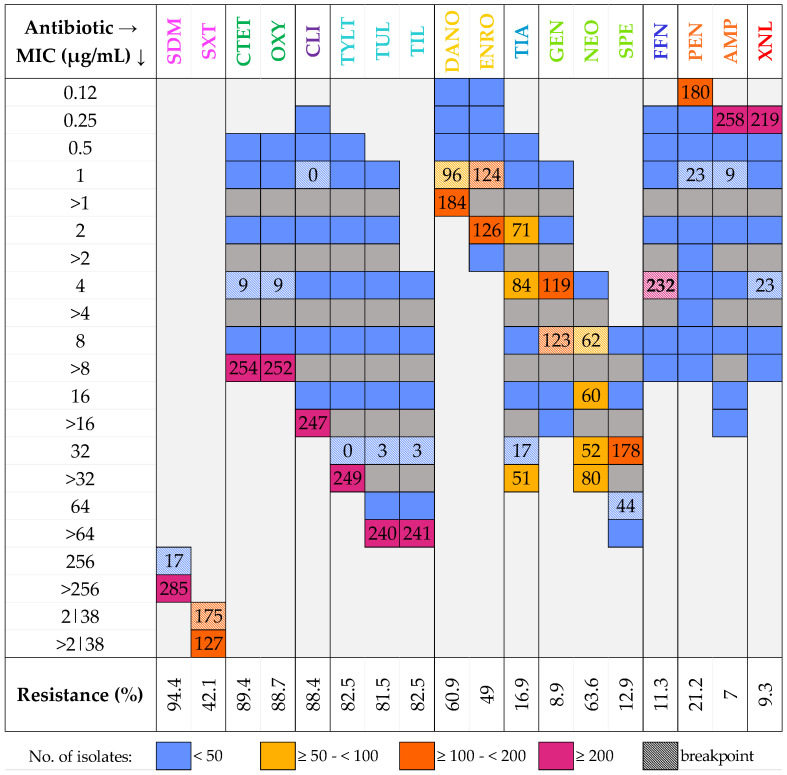
MIC distribution heatmap for tested antimicrobials and resistance percentages in 302 isolates, displayed at breakpoint concentrations and MIC values where the isolate count is ≥50. Same-family antimicrobials (e.g., SXT, SDM) share color tones. Grey boxes indicate non-applicable MICs.

**Figure 3 vetsci-11-00040-f003:**
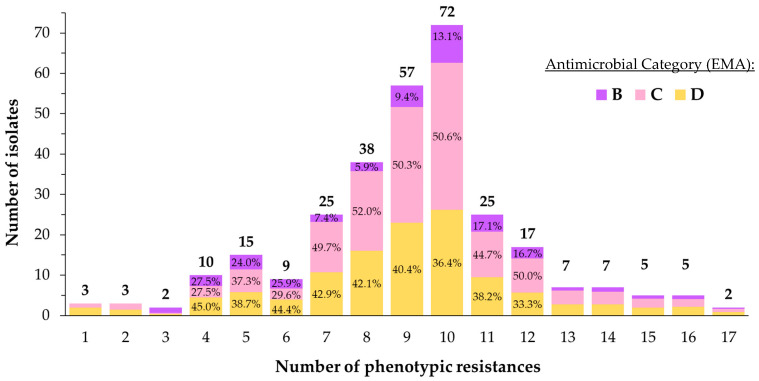
Distribution of number of phenotypic resistances in isolates. Distribution of resistance percentages according to EMA [22] classification of studied antimicrobials is also shown.

**Figure 4 vetsci-11-00040-f004:**
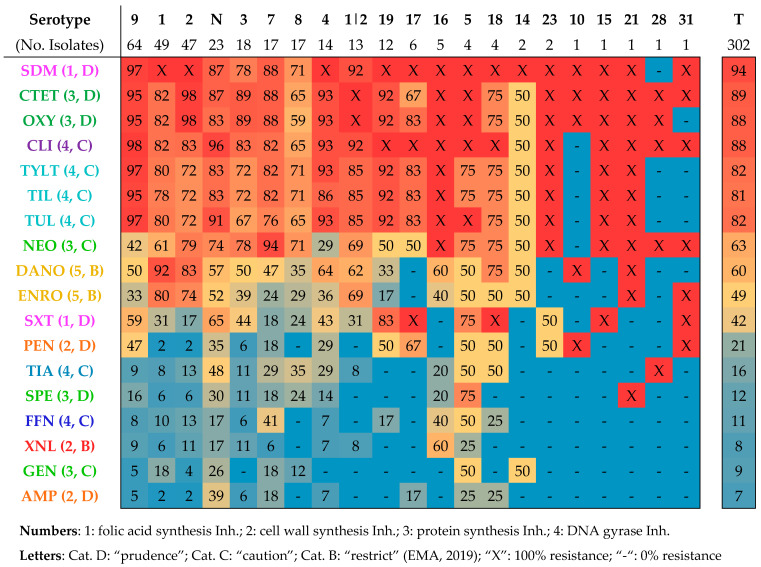
Serotype-specific antimicrobial resistance heatmap, expressed with percentages. Warmer colours indicate higher levels of AMR, while cooler tones denote lower resistance percentages. Non-typable isolates (N) are also included. Total isolates with colour gradients depending on AMR percentages (right) are shown for reference. Antimicrobial use category according to EMA [26] (B, C, or D) and antimicrobial mechanism of action (numbers 1 to 5) are also displayed.

**Figure 5 vetsci-11-00040-f005:**
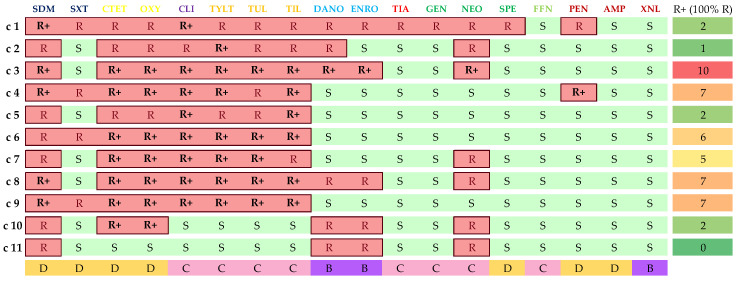
Phenotypic AMR distribution within the clusters, depending on the most frequent observations. Results were classified as sensitive (S), resistant (R), or “R+” for 100% resistance. EMA antibiotic categorization [26]—D, C, or B (represented with different colours)—is also included for each antimicrobial.

**Figure 6 vetsci-11-00040-f006:**
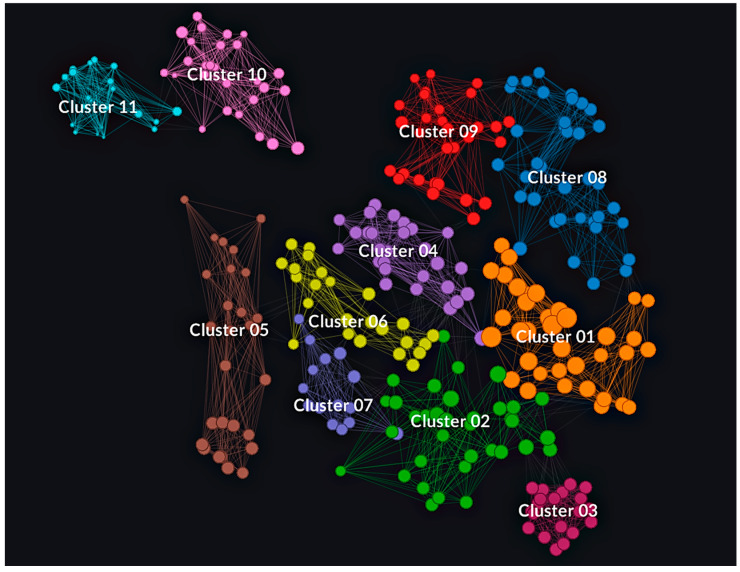
AMR-based cluster representation of the 302 isolates using Graphext. Nodes with bigger sizes indicate higher levels of AMR, while nodes with smaller sizes indicate higher sensitivity levels to the tested antimicrobials. The connections among these nodes illustrate the relationships between isolates within the same cluster.

**Figure 7 vetsci-11-00040-f007:**
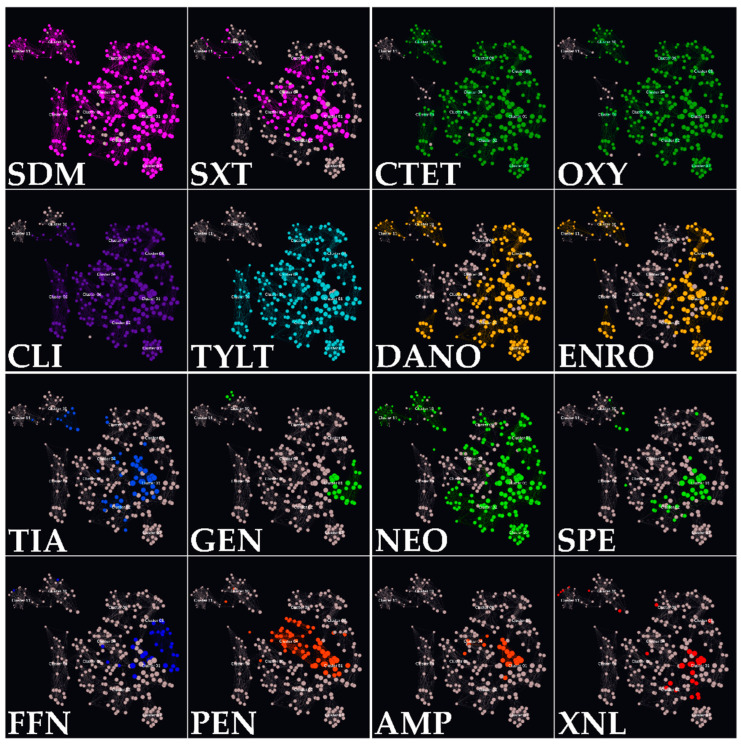
Graphext representation of the AMR (coloured nodes) detected among the isolates. Resistances to antimicrobials of the same family are represented with identical colours. Macrolides TIL and TUL, which shared an AMR pattern nearly identical to TYLT, are not represented.

**Figure 8 vetsci-11-00040-f008:**
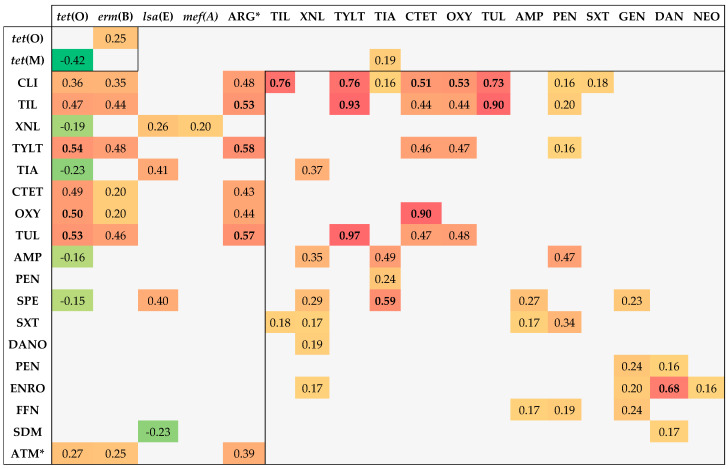
Phi coefficients for the significant associations (*p* < 0.05) among the analysed variables. ARG–ARG associations are positioned in the top left corner, whereas ATM–ATM associations are in the bottom right. ARG–ATM associations are situated between them. Warm colours indicate positive phi coefficients, while negative coefficients are represented with green tones. The abbreviations with asterisks (*) refer to the number of phenotypic resistances (ATM) and the number of AMR genes (ARG).

**Table 1 vetsci-11-00040-t001:** Antimicrobials included in the Sensititre™ BOPO6F plates (Thermo Fisher Scientific, Madrid, Spain), their concentration ranges, and CLSI [24] and EUCAST [25] breakpoints.

Antimicrobial Family	Antimicrobial	Range (µg/mL)	BP (µg/mL)
Aminoglycosides	Spectinomycin (SPE)	8–64	64
	Gentamicin (GEN)	1–16	8
	Neomycin (NEO)	4–32	8
Cephalosporins	Ceftiofur (XNL)	0.25–8	4
Amphenicols	Florfenicol (FFN)	0.25–8	4
Lincosamides	Clindamycin (CLI)	0.25–16	1
Macrolides	Tylosin tartrate (TYLT)	0.5–32	32
	Tilmicosin (TIL)	4–64	32
	Tulathromycin (TUL)	1–64	32
Penicillins	Ampicillin (AMP)	0.25–16	1
	Penicillin (PEN)	0.12–8	1
Pleuromutilins	Tiamulin (TIA)	0.5–32	32
Quinolones	Danofloxacin (DANO)	0.12–1	1
	Enrofloxacin (ENRO)	0.12–2	1
Sulfonamides	Sulfadimethoxine (SDM)	256	256
	Trimethoprim-sulfamethoxazole (SXT)	2|38	2|38
Tetracyclines	Chlortetracycline (CTET)	0.5–8	4
	Oxytetracycline (OXY)	0.5–8	4

**Table 2 vetsci-11-00040-t002:** *Streptococcus suis* serotype distribution by round of PCR amplification.

Serotype	R.O.A. *^1^	N° of Isolates	% of Isolates	% R.O.A.
1	First *^2^	49	16.2%	71.2%
14	First *^2^	2	0.7%	
2	First *^2^	47	15.6%	
1/2	First *^2^	13	4.3%	
3	First	18	6.0%	
7	First	17	5.6%	
9	First	64	21.2%	
16	First	5	1.7%	
4	Second	14	4.6%	16.9%
5	Second	4	1.3%	
8	Second	17	5.6%	
18	Second	4	1.3%	
19	Second	12	4.0%	
10	Third	1	0.3%	3.6%
15	Third	1	0.3%	
17	Third	6	2.0%	
23	Third	2	0.7%	
31	Third	1	0.3%	
21	Fourth	1	0.3%	0.7%
28	Fourth	1	0.3%	
NT	-	23	7.6%	-

*^1^ R.O.A.: round of amplification (each round included different primer sets); *^2^ Distinction of serotypes was achieved employing primers targeting the *cpsK* gene.

**Table 3 vetsci-11-00040-t003:** Distribution of *S. suis* serotypes isolated from different anatomic locations.

Serotype	C.N.S.	Lungs	Joints	Other *	Total
9	34	16	13	1	64
1	20	9	19	1	49
2	20	14	9	4	47
3	2	15	0	1	18
7	7	3	4	3	17
8	1	14	1	1	17
4	5	7	1	1	14
1|2	10	1	2	0	13
19	8	2	1	1	12
OS *	11	11	6	0	28
NT *	10	7	4	2	23
Total	128	99	60	15	302

* Other: other anatomic isolation sites; OS: other serotypes; NT: non-typable isolates.

**Table 4 vetsci-11-00040-t004:** Antimicrobial resistance patterns among the studied *S. suis* isolates.

Serotype	Antimicrobial Resistance Pattern	Isolates (%)
1	*erm*(B)*+|mef*(A/E)*−|tet*(M)*−|tet*(O)*+|lsa*(E)*−|lnu*(B)*−*	61.2%
	*erm*(B)*−|mef*(A/E)*−|tet*(M)*−|tet*(O)−*|lsa*(E)*−|lnu*(B)*−*	16.3%
	*erm*(B)*−|mef*(A/E)*−|tet*(M)*−|tet*(O)*+|lsa*(E)*−|lnu*(B)*−*	14.3%
	Other antimicrobial resistance patterns	8.2%
2	*erm*(B)*−|mef*(A/E)*−|tet*(M)*−|tet*(O)*+|lsa*(E)*−|lnu*(B)*−*	48.9%
	*erm*(B)*+|mef*(A/E)*−|tet*(M)*−|tet*(O)*+|lsa*(E)*−|lnu*(B)*−*	36.2%
	*erm*(B)*−|mef*(A/E)*−|tet*(M)*+|tet*(O)*−|lsa*(E)*−|lnu*(B)*−*	8.5%
	Other antimicrobial resistance patterns	6.4%
9	*erm*(B)*+|mef*(A/E)*−|tet*(M)*−|tet*(O)*+|lsa*(E)*−|lnu*(B)*−*	73.8%
	*erm*(B)*−|mef*(A/E)*−|tet*(M)*−|tet*(O)*+|lsa*(E)*−|lnu*(B)*−*	13.8%
	*erm*(B)*+|mef*(A/E)*−|tet*(M)*−|tet*(O)*−|lsa*(E)*−|lnu*(B)*−*	7.7%
	Other antimicrobial resistance patterns	4.6%
OS * (including 3, 4, 7, 8, 1|2, and 19)	*erm*(B)*+|mef*(A/E)*−|tet*(M)*−|tet*(O)*+|lsa*(E)*−|lnu*(B)*−*	36.1%
	*erm*(B)*−|mef*(A/E)*−|tet*(M)*−|tet*(O)*+|lsa*(E)*−|lnu*(B)*−*	32.8%
	*erm*(B)*−|mef*(A/E)*−|tet*(M)−*|tet*(O)*−|lsa*(E)*−|lnu*(B)*−*	12.6%
	Other antimicrobial resistance patterns	18.5%
NT *	*erm*(B)*−|mef*(A/E)*−|tet*(M)*−|tet*(O)*+|lsa*(E)*−|lnu*(B)*−*	26.1%
	*erm*(B)*−|mef*(A/E)*−|tet*(M)*−|tet*(O)*−|lsa*(E)*−|lnu*(B)*−*	21.7%
	*erm*(B)*+|mef*(A/E)*−|tet*(M)*−|tet*(O)*+|lsa*(E)*−|lnu*(B)*−*	13.0%
	Other antimicrobial resistance patterns	39.1%
Total	*erm*(B)*+|mef*(A/E)−*|tet*(M)*−|tet*(O)*+|lsa*(E)−*|lnu*(B)*−*	46.5%
	*erm*(B)−*|mef*(A/E)*−|tet*(M)*−|tet*(O)*+|lsa*(E)*−|lnu*(B)*−*	27.7%
	*erm*(B)*−|mef*(A/E)*−|tet*(M)*−|tet*(O)*−|lsa*(E)*−|lnu*(B)*−*	9.9%
	Other antimicrobial resistance patterns	15.8%

* OS: serotypes other than 1, 2, or 9. In this table, it also refers to serotypes 3, 4, 7, 8, 1|2, and 19. * NT: serotypes not typable by multiplex PCR.

**Table 5 vetsci-11-00040-t005:** Main characteristics of the AMR clusters. Presence (+) or absence (−) of the genes via PCR detection is represented with the corresponding symbols. The last column includes both the number of phenotypic resistances and the resistance interval within the cluster.

AMR Cluster	No. of Isolates	Main Two Serotypes	Main Isolat. Site	*erm*(B)	*mef*(A/E)	*tet*(M)	*tet*(O)	*lsa*(E)/ *lnu*(B)	Phenotypic AMRs
c1	36	9, NT	CNS	+	−	−	+	−	12 (10–17)
c2	35	1, 8	Lungs	+	−	−	+	−	11 (7–13)
c3	20	2, 1	CNS	+	−	−	+	−	10
c4	28	9, 4	CNS	+	−	−	+	−	10 (8–12)
c5	28	9, 1	CNS	+	−	−	+	−	9 (4–9)
c6	24	9, 1	CNS	+	−	−	+	−	8 (7–10)
c7	15	3, 7	Lungs	+	−	−	+	−	8 (7–9)
c8	37	2, 1	CNS	−	−	−	+	−	10 (9–13)
c9	30	2, 19	CNS	−	−	−	+	−	9 (7–11)
c10	27	2, 3	Lungs	−	−	−	+	−	7 (4–9)
c11	23	1, 8	Lungs	−	−	−	−	−	4 (1–7)

## Data Availability

To ensure the privacy of the data concerning the laboratories and farms involved, individual data confidentiality is maintained. As a result, specific details about these establishments cannot be publicly disclosed.

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
