# Peer review of "Streptococcus suis Research Update: Serotype Prevalence and Antimicrobial Resistance Distribution in Swine Isolates Recovered in Spain from 2020 to 2022"

_vetsci, 2024, doi:10.3390/vetsci11010040_

Round 1

Reviewer 1 Report

Comments and Suggestions for Authors

The paper is well written. The methodology used is appropriate. The results are completely described and the discussion is complete.

Only few suggestions:

Line 41 Delete "virulence gene" (there is no mention about it in the text)

Table 1: Complete the word "Antimicrob." with "Antimicrobial"

Table 4: include a footnote for "OS" and "NT" explanation

Author Response

Dear reviewer, please see the attachment:

Reviewer 2 Report

Comments and Suggestions for Authors The topic studied really requires research, because S. suis is one of the
most pathogenic bacteria that causes losses in the development of the
piglets, and millions of deaths in the swine industry.
Research like this must be done not only in Spain, but also in other
countries where pig farming is an important economic activity.
Furthermore, there is a risk of this bacterium infecting humans
(zoonosis).
The indiscriminate use of antibiotics in pigs occurs due to the absence of
effective commercial vaccines. As this bacterium colonizes piglets early,
pig farmers apply antimicrobials during the perinatal phase in order to
control the disease, and this practice causes multiple resistance to
antimicrobials, which was demonstrated in the present study.
Really, it is necessary to adopt preventive measures, by restricting
indiscriminate antimicrobial usage. The authors used advanced
laboratory techniques to carry out the analyses in 302 isolates, when
there were identified the prevalent serotypes, genotypic and phenotypic
antimicrobial resistance features, and associations between 30 resistance
genes and phenotypic resistances. Besides to find the most prevalent
Serotypes: 9 (21.2%), 1 (16.2%), 2 (15.6%), 3 (6%), and 7 31 (5.6%),
the antimicrobial resistance genes, including tet(O), erm(B), lsa(E) and
lnu(B), tet(M), and 33 mef(A/E), were analysed, which were present in
85.8%, 65.2%, 7%, 6.3%, and 1% of the samples, respectively.
Susceptibility testing for 18 antimicrobials revealed high resistance
levels, particularly 35 for clindamycin (88.4%), chlortetracycline (89.4%
), and sulfadimethoxine (94.4%), and seven significant associations
(p < 0.0001) were detected, correlating specific antimicrobial resistance
genes to the observed phenotypic resistance.
The statistical evaluation, data tabulation and graphs are clear.
An in-depth discussion allows the reader to understand the results and
thus reach a conclusion.
The findings contributed to know the most prevalent S. suis serotype
distribution and its antibiotic resistance profiles in Spain.

Author Response

(The authors gave the same response as above.)

Reviewer 3 Report

Comments and Suggestions for Authors

Dear Authors, 

Congrats for your work. I am sorry to say the paper still needs more work to be ready for publication. However it has the work needed to be apt, though some bits need to be fixed. 

Line 13. I think it is important to specify if they are clinical or not or a mixture, even on the abstract.

Line 50. The PRDV is not age restricted. 16 and 20 weeks are not included on the definition of the complex. Please, remove.

Line 51. Key ones as SIV, PCV2, and APP are missing in your list. While PRCV is not so important.

Line 57. What do you mean with primary? S. suis acts as secondary or opportunistic in too many occasions post-weaning. While it is clarly primary on lactations. I would recommend removing or modify – or at least explain- the word primary.

Line 58. Also skin.

Line 66. The word commonly is difficult to interpret. Common signs, common occurrence ? I would say none of them cause it does not commonly cause all those signs. Some of them belong to an acute problem, some others to a chronic, you rarely find all of them on the same pig. And again, S.suis is not so frequent. There are reason why it happens where it happens.

Line 69. 29 serotypes is something debatable. I would use a more renown reference and provide a range rather than a number as world experts present other figures. The reference you used and not renowned authors.

Line 68. The serotype classification based on cell surface antigens is based on serology. The serotype classification based on genes encoding the proteins enabling the synthesis of the surface antigens are done by PCR. You mixed your sentence making it sound wrong. Please be accurate and clear. The PCR does not detect the surface antigens. It detects the genes independently if they were expressed. So some non-typeable strains and strains that switch off the capsule can still be typed by PCR.

LINe 89, what do you mean for universal. It is not clear, better to state there is not vaccine for every serotype… as it happens for a large number of diseases.

LINE 87. This sentence would require a reference.

LINE 91. The perinatal antimicrobials are prophylactic, not metaphylactic cause disease takes to express several days or weeks. It is used in every litter and not every litter needs it. Please, do not use metaphylactic there, it is not true.  

LINE 98. Sorry but S.suis is not one of the most menacing pathognes on the swine industry. It is important, but not menacing as it does not spread in a epidemic way. Otherwise we would already have better tools to fight it. Sorry but it is not. Please, remove, it is an exaggeration, even if there is a reference, we can let it perpetuate.

LINE 99. Please state if that is for pigs or humans.

LINE 109. Please, any info if they were clinical or non-clinical samples? Please, could you state how many colonies where isolated per sample? It is frequent to find several strains on a single sample, specially on non-clinical ones.

Figure 3 and others. Please add abbreviation of antimicrobials on the foot text. Also I would recommend to use a much standard format for presenting the MIC distribution as you can find on other literature about AMR. And example

https://www.ncbi.nlm.nih.gov/pmc/articles/PMC8944821/

https://www.ncbi.nlm.nih.gov/pmc/articles/PMC5548070/

https://www.ncbi.nlm.nih.gov/pmc/articles/PMC6318959/

https://www.ncbi.nlm.nih.gov/pmc/articles/PMC10512786/

I personally recommend creating charts that can still be useful when printed B/W but that is just a personal opinion. In this chart you put colour-blind-unfriendly colours where perfectly fits a number.

Same comments for the other charts.

Another comment in general about the literature used. Out of the 6 main S.suis research teams in the world, you only used literature from two of them (Gottschalk-CAN and Brokmeier-USA) but you do not include nothing about V. Aragon (UAB-Spain), AW. Tucker (Cambridge, UK), Wren – (Wagenigen, NED) and Rui Zhou (Wuhan, China). They could be very helpful to have a more critical understanding of the effect of the sample set on the results, as well as other views about serotyping, ST, PCR and antimicrobial resistance patters, or MIC phenotype compared with genotype. Some of them are published in nature comm. This one is not from nature but is quite similar to your work in some aspects. https://www.ncbi.nlm.nih.gov/pmc/articles/PMC8422772/. I would recommend to have a look on that literature and use is to compare with your data and build the discussion unless you preferred to excluded it on purpose. Some of the literature you use for comparison is 20 years old or Asiatic while you do not included European material from the last 10-15 years. I think that is the material to compare your samples with in a first place as comparison with other continents where they still use other products in a different way is somehow a not fair comparison, and actually not very valuable.

There are also descriptions on the literature as some of these AMR genes are placed too close to each other in the genome and very possibly co-selected.

LINE 276. To improve the results interpretation, please, could you specify which antimicrobial tested belong to each list.

LINE 448. I would refrain to use the word comprehensive when you do not really state how many farms the samples come from, age of pigs, or the territorial representativity, its clinical relevance… So, it can be wide, but you have not provided enough information to defend the word comprehensive.  

LINE 453. ST9 may be the most predominant in postweaning cases but not as much in preweaning cases. It is important to mention cause they are two completely different type of problems. Also, your sampleset is biased by samples that were kept or sent by a reason. Then you do not know if 9 is the more prefdominant, or the more involved in an specific type of problem. So I would rephrase carefully to express just what 9 is. The most predominant in your collection, but we do not know what is your collection and how it was built-up, clinical, not clinical, pre or post weaning…

Line 458. Please could you specify if it is transmission from pigs to human or infection in humans in contacts with pig products or subproducts, or wild-boars. .

LINE 466 please rephrase, I do not get the point of this sentence. If you mean

Line 478. Use of English issue. The sentence reads as zoonosis are frequent. These serotypes may be the most common on the rarely reported cases of zoonosis.

Line 540 to 570. You make a lot of comparison with a few papers. It breaks the flow of discussion and loss the reader. It would quite positive to have a table with this comparison of different sources, similar to the “ R I S “ tables on the references above.

LINE 598. Florfenicol has been used till quite recently and is still used in EU including Spain.  Resistances usually apply for the whole group.

Line 662  - you found association in between the level of the resistances in your cluster with good p-value, but you need to ensure that your samples are not very epidemiologically related because that could have been a bias increases your p-values. Please, add mote info about the description of the samples to avoid critics in that respect.

Author Response

(The authors gave the same response as above.)

Reviewer 4 Report

Comments and Suggestions for Authors

1.     Line 14 “were 9” change to “were serotype 9)

2.     Delete the sentence line 24-25 “Studying S. suis resistance to antimicrobials is crucial for addressing potential threats to public health, as it helps identify effective strategies to prevent the spread of antibiotic-resistant bacteria.” Since the similar meaning already mentioned in the previous sentence.

3.     change the sentence line 39-40 “…..resistance profiles in Spain. This research offers valuable insights for veterinary and public health efforts in the management of S. suis infections” to “…..resistance profiles in Spain, offer valuable insights for veterinary and public health efforts in the management of S. suis infections.”

4.     In Table 5, the vertical lines between columns should be deleted.

Author Response

(The authors gave the same response as above.)
